# Novel Silyl Ether-Based Acid-Cleavable Antibody-MMAE Conjugates with Appropriate Stability and Efficacy

**DOI:** 10.3390/cancers11070957

**Published:** 2019-07-08

**Authors:** Yanming Wang, Shiyong Fan, Dian Xiao, Fei Xie, Wei Li, Wu Zhong, Xinbo Zhou

**Affiliations:** National Engineering Research Center for the Emergency Drug, Beijing Institute of Pharmacology and Toxicology, Beijing 100850, China

**Keywords:** antibody-drug conjugate, linker, acid-cleavable, silyl ether, monomethyl auristatin E

## Abstract

Antibody-drug conjugate (ADC) is a novel efficient drug delivery system that has been successfully used in clinical practice, and it has become a research hotspot in the anti-tumor drug field. Acid-cleavable linkers were first used in clinical ADCs, but their structural variety (e.g., hydrazone and carbonate) is still limited, and their stability is usually insufficient. Designing novel acid-cleavable linkers for the conjugation of the popular cytotoxin monomethyl auristatin E (MMAE) has always been a significant topic. In this paper, we generate a novel, silyl ether-based acid-cleavable antibody-MMAE conjugate, which skillfully achieves efficient combination of amino-conjugated MMAE with the acid-triggered silyl ether group by introducing *p*-hydroxybenzyl alcohol (PHB). The stability, acid-dependence cleavage, effective mechanism, efficacy and safety of the resulting ADC were systematically studied; the results show that it exhibits a significant improvement in stability, while maintaining appropriate efficacy and controlled therapeutic toxicity. This strategy is expected to expand a new type of acid-cleavable linkers for the development of ADCs with highly potent payloads.

## 1. Introduction

Antibody-drug conjugate (ADC), comprised of an antibody, a cytotoxin and an appropriate linker [1,2], achieves efficient delivery of a cytotoxin by a monoclonal antibody, greatly enhancing the therapeutic index of cytotoxic drug and potentially reducing systemic toxicity [3,4,5]. Recently, ADCs have shown great promise in the treatment of various types of cancer [6,7], with more than 80 ADCs in different stages of clinical development [8].

There are many important factors involved in the successful development of ADC, such as the selection of suitable targets and antibodies, efficient cytotoxins, and appropriate linkers [9,10,11]. Among them, the linker is one of the core factors for the success of the ADC design, which needs to maintain the stability of the ADC in systemic circulation and release the cytotoxin after internalization at the target site [12]. Based on the release mechanism, linkers are generally divided between acid-cleavable linkers, reduction-cleavable linkers, enzyme-cleavable linkers, and non-cleavable linkers [13,14].

Acid-cleavable linker was the first to be used in early ADC constructs, which cleverly utilized the pH difference between tumor tissue (4.0–5.0) and plasma (~7.4) [14,15]. The hydrazone-based acid-cleavable linkers (Figure 1A,B) are not only the first ones that were successfully used in ADCs, but they were also used in two of the four currently approved ADCs [16]. Additionally, a carbonate-based acid-cleavable linker has been tried to conjugate cytotoxin SN-38 (Figure 1C) [17]; for example, a phase III trial for the resulting sacituzumab govitecan has been initiated, and it has earned the Breakthrough Therapy designation from the U.S. Food and Drug Administration (FDA) [18]. However, the lack of stability of current acid-cleavable linkers could potentially lead to toxicity during clinical applications [14,19]. The first approved ADC, Mylotarg^®^ (Pfizer Inc., Brooklyn, NY, USA), was voluntarily withdrawn in 2010 mainly because of the instability of the hydrazone linker, and it still has dose-limiting hepatotoxicity, although it was subsequently re-marketed [20,21]. It is also generally believed that existing acid-cleavable linkers are not suitable for conjugating highly cytotoxic drugs, just as IMMU-132 improved efficacy by improving drug/antibody ratio (DAR) to 7.6 due to the insufficient effectiveness of SN-38 (IC_50_ = 10^−7^–10^−8^ mol/L). However, high DAR values might lead to faster clearance or potential immunogenic reactions in some cases [22,23]. Overall, the structural category of acid-cleavable linkers is still limited, and insufficient stability greatly restricts its wide application in constructing ADCs.

For the most popular, highly cytotoxic, monomethyl auristatin E (MMAE), which accounts for nearly one-third of the clinical ADCs [18], it has always been designed to employ enzyme-cleavable linkers, such as the commonly used Val-Cit (VC)-based cathepsin B-cleavable linker [24,25,26,27,28]. Nevertheless, due to susceptibility to an extracellular carboxylesterase [29], a number of ADCs failed during development due to tolerability reasons [30,31]. Overall, the current linker design for ADCs is not yet perfect, and the design of acid-cleavable linkers for the conjugation of MMAE has always been a topic worth exploring.

In this paper, a novel, silyl ether-based acid-cleavable ADC was generated with MMAE as the potent payload. Methylsilyl ether groups have characteristics that are sensitive to acidity, and have been used as protective bases for organic synthesis and drug release groups for nanomaterials [32,33,34]. In addition, these groups have also been used in the design of prodrugs, mainly limited to hydroxyl-conjugated drugs (e.g., acetaminophen, and docetaxel) due to the limitation of the silyl ether structure [35,36]. Although this type of linker has been tried for model ADC construction, which exhibits the characteristics of stability and acid-dependent cleavage, gemcitabine does not meet the high-efficiency requirements for ADC’s payload and it has not been evaluated for its efficacy [37]. By introducing *p*-hydroxybenzyl alcohol (PHB), the ADC skillfully achieves the efficient combination of amino-conjugated MMAE with acid-triggered silyl ether groups (Figure 1D). The resulting ADC exhibits improved stability, efficient release of MMAE just like enzyme-cleavable ADCs, the appropriate efficacy, and controlled toxicity. This linker system is expected to deepen the cognition of acid-cleavable linkers and provide a novel conjugating model for MMAE. As an addition to the existing acid-cleavable linkers, such as hydrazone and carbonate, this strategy may provide an alternative opition for the development of ADCs.

## 2. Results

### 2.1. Design and Synthesis of Silyl Ether-Based ADC Payload

In the paper, we have attempted to introduce a novel, acid-cleavable mechanism for the design of ADC with MMAE as the potent payload. We designed the target ADCs using silyl ether groups that have good acid-cleavable properties. Additionally, inserting PHB between MMAE and the silyl ether group achieved an efficient combination of amino-containing cytotoxin. Various substituted silyl ethers were designed as new linkers, and evaluated for their stability and acid-cleavable properties. However, these silyl ethers-based linkers were found to have different stabilities during the synthesis process. The dimethyl, diethyl and diphenyl substitution modes were not feasible due to their inherent instability (Appendix A). In addition, larger alkyl substituents will undoubtedly increase the hydrophobicity of the linker, which is detrimental to the properties of the ADC. Thus, a diisopropyl-silyl ether-based linker was prepared (Figure 2), and further used for the conjugation of MMAE to an anti-human epithelial growth factor receptor 2 (HER2) antibody to develop a novel acid-cleavable ADC.

### 2.2. Antibody Conjugation

The reaction of maleimide compound with antibody thiols was evaluated using a humanized anti-HER2 antibody mil40, which is a biosimilar of trastuzumab. The cytotoxin payload was typically conjugated to the antibody through cysteine, obtained from the reduction of inter-molecule chain disulfide bonds. By controlling the equivalent of the reducing agent tris(2-carboxyethyl)phosphine hydrochloride (TCEP), a defined number of antibody payload modifications can be achieved, which were in a relatively fixed position. Hydrophobic interaction chromatography (HIC) analysis of the final ADC product allowed for resolution of the conjugates into several major peaks with an average DAR of about 5.5 (Appendix A) [38].

### 2.3. The Stability Assays of the Conjugate in Plasma

The silyl ether-based linker-MMAE conjugate and the prepared ADC (mil40-6) was initially evaluated for their in vitro stability in human plasma. In this experiment, the ADC of DAR ~5.5 was selected, and the silyl ether-based linker-MMAE conjugate was added to a solution of excess *N*-acetyl-*L*-cysteine (NAC) to complete conversion to NAC-linker-MMAE (NAC-6) before test [38]. The above mixtures were incubated in 50% plasma at 37 °C for a period of 7 days. Aliquots were taken at the indicated time points and analyzed by LC-MS/MS for the release of free cytotoxin MMAE. As shown in Figure 3, less than 3% of the total MMAE in the NAC-linker-MMAE conjugate was released during 7 days in human plasma. Correspondingly, about 24% of the total MMAE in ADC was off-targeted. Overall, this type of acid-cleavable linker system displayed acceptable in vitro plasma stability for the initial study.

### 2.4. Effect of Acidity on ADC Stability

Further, the pH-dependence of the cytotoxin release process of the silyl ether-based acid-cleavable ADC (mil40-6) was evaluated. Under three selected pH values (pH 7.4, 5.5, and 4.5), this novel ADC (DAR ~ 5.5) was incubated for 7 days at 37 °C. As showed in Figure 4, the release rate of MMAE exhibited significant pH-dependence, with approximately one-third of the total MMAE in ADC being released after 7 days under neutral conditions (pH = 7.4), whereas the loaded MMAE rarely been detected under acidic test conditions (pH = 4.5). This result demonstrated that the diisopropyl-substituted silyl ether has a certain acid-dependent acid-cleavable property in this model [37].

### 2.5. In Vitro Potency Assay

The prepared ADC with an average DAR of 5.5 was evaluated for its cytotoxicity against HER2-positive cell lines (NCI-N87, BT-474, and MDA-MB-453) [39], and the HER2-negative cell lines (MCF-7 and MDA-MB-231) [40], along with the naked antibody (mil40) and cytotoxin (MMAE) as controls [41]. The ADC displayed excellent anti-tumor activity against all HER2-positive cell lines (IC_50_, 0.028–0.170 nM), whereas exhibited low activity against HER2-negative cell lines (IC_50_, 7.742 ~ >1000 nM). The ADC exhibited a similar maximum inhibition rate to naked antibody in BT-474 cells, mainly because the antibody itself was very sensitive to BT-474 cells just as we reported previously [38]. As a comparison, MMAE showed potent anti-tumor activity in both HER2-positive and -negative cell lines with almost no antigenic selectivity (Table 1).

### 2.6. Trafficking Assay by Fluorescence Microscopy

To release cytotoxin inside cancer cells, ADC needs to be internalized and exposured in the lysosomal compartments. Thus, the internalization and trafficking of the ADC mil40-6 was detected via fluorescently labeled antibody in HER2^+^ BT-474 cell line with a laser scanning confocal microscope. Lysosomal compartments were visualized by staining with lysosomal-associated membrane protein 1 (LAMP-1). As shown in Figure 5, the mil40 and mil40-6 were incubated with the lysosomal labeled BT-474 cells at 4 °C, under which the antibody was not been internalized (no obvious red). However, after incubation at 37 °C for 16 h, the intracellular antibody signals colocalized with the signals of LAMP-1 (green), indicating that the mil40-6 (red) can be efficiently internalized in cancer cells through endocytosis (Pearson’s correlation = 0.467; Mander’s overlap = 0.567), which is very similar to the naked antibody mil40 (Pearson’s correlation = 0.538; Mander’s overlap = 0.525).

### 2.7. Linker Composition on Inhibition of Microtubule Polymerization

As an auxiliary verification and interpretation of the in vitro cytotoxicity, an in vitro microtubule polymerization assay was performed to confirm that the incorporated structural variations and elaborations to MMAE did not impair the drug’s ability to inhibit microtubule polymerization (Figure 6). The MMAE and corresponding Cys-linker-MMAE conjugate (Cys-6) were used at 3 μM and 0.3 μM.

The conjugate and free MMAE showed similar inhibition rates of tubulin polymerization at various concentrations. As expected, the panel of drug/linkers resulted in microtubule polymerization inhibition similar to unmodified cytotoxin MMAE. This result may provide some basis for the high in vitro cytotoxicity of this novel ADC.

### 2.8. Cell Cycle Effects

The anti-tumor mechanisms of the acid-cleavable ADC was further evaluated with respect to cell cycle arrest. MMAE is a synthetic analogue of the natural product dolastatin 10, which can bind to tubulin and block cell division by inhibiting microtubule assembly [28,42,43]. Normally, ADCs consisting of MMAE induce G_2_/M phase arrest in target cells preceding the onset of apoptosis [44,45]. As showed in Figure 7, the in vitro effects of mil40-6 (10 ng/mL, 50 ng/mL, and 100 ng/mL) on BT-474 cell cycle arrest after 6 days’ treatment were presented. Compared to the vehicle, the ADC showed no cell cycle inhibition at the low dose group (10 ng/mL). While, few cells treated with mil40-6 remained in the G_0_/G_1_ phase when the dose is increased, with most were either in the S or G_2_/M phases. At the dose of 100 ng/mL, the proportion of G_2_/M phase in ADC-treated cells was 34.04% higher than vehicle, indicating that this ADC can suppress the cell cycle to the G_2_/M phase.

### 2.9. In Vivo Potency in Xenografted Nude Mice

To evaluate the efficacy of the acid-cleavable ADC (mil40-6) in vivo, we established NCI-N87 xenografts in nude mice and then treated these BALB/c mice with the naked antibody (mil40), ADC, and vehicle [46]. As shown in Figure 8A, the animals were given vehicle, mil40, and ADC on days 0, 7, 14, and 21, and 4 intravenous injections of ADC at doses of 2.5 and 5 mg/kg. Both doses of the ADC treatment group produced significant and sustained anti-tumor effects and showed significant dose-efficacy dependence. As a comparison, 4 intravenous injections of mil40 at doses of 5 mg/kg resulted in almost no tumor suppression. Compared with mil40, the ADC had a significant increase in potency with the same dose (*p* = 0.0054), with a 93.6% tumor inhibition rate.

During the treatment period, there was no weight loss observed in all of the treatment groups. Body weight of the test mice was not significantly different between experimental and control groups, indicating that the ADC and naked antibody were all preliminarily well-tolerated at the therapeutic dose (Figure 8B).

### 2.10. Hematology Safety Study

To further verify the safety of the silyl ether-based acid-cleaving ADC (mil40-6), changes in hematology between vehicle and treatment groups (ADC at doses of 5 mg/kg) were compared at the end of administration (one week after the end of administration; day 28).

The unpaired two-tailed *t* tests for the hematological parameters of the blood samples showed that there was no significant difference (*p* < 0.05) between the ADC treatment group (5 mg/kg) and the vehicle (white blood cells: *p* = 0.6083; red blood cells: *p* = 0.1655; hemoglobin: *p* = 0.8078; platelets: *p* = 0.5633; neutrophils: *p* = 0.8072; lymphocytes: *p* = 0.5030; monocyte: *p* = 0.9230; eosinophils: *p* = 0.2161; large unstained cell: *p* = 0.4772), with the hematological parameters of the test animals shown in Figure 9. The above results demonstrated that the ADC did not produce significant hematological toxicity at therapeutic doses.

## 3. Discussion

The design of the linker is one of the core elements for constructing ADCs, as it has a direct and significant impact on stability, toxicity and pharmacokinetics [47,48]. Linkers with different drug release mechanisms currently need to constantly balance the relationship between efficacy and toxicity of ADCs to obtain better clinical benefits, making the developing of diverse types of linkers a historically attractive proposal.

The MMAE (IC_50_ ~ 10^−11^ mol/L) developed by Seattle Genetics (Bothell, WA, USA) is the most widely used cytotoxin in current ADCs [18,49]. However, the conjugating mode of MMAE-based ADCs is still relatively simple, as the high toxicity of MMAE generally requires the linker to have sufficient plasma stability to reduce off-target toxicity. Currently, MMAE mostly uses a Val-Cit (VC)-based cathepsin B-cleavable linker to achieve conjugation with antibodies; this enzymatic linker system has good stability, but its structural complexity may lead to additional tolerance risks. It is generally believed that MMAE is not suitable for existing acid-cleavable linkers that are based on hydrazone, carbonate acid, or others due to its high toxicity; these types of linkers are currently only used for conjugating cytotoxins with relatively lower activities, such as doxorubicin, calicheamicin, and SN-38. These acid-cleavable ADCs usually require increased doses, frequency of dosing, or DAR to ensure efficacy, which may result in dose-limiting toxicity and immunogenic risk [50].

In this study, we generated a novel MMAE-based acid-cleavable ADC with relatively high plasma stability. It has been reported that silyl ether groups are sensitive to acidity and their cleavage can be precisely regulated by adjusting the size of the alkyl substituents in their structure [33,34,35,51]. However, these groups have only been used in the design of prodrugs for hydroxyl-containing drugs due to the limitation of the silyl ether structure. By introducing PHB, the ADC skillfully achieves an efficient combination of amino-conjugated MMAE with acid-triggered silyl ether groups. In addition, the further introduction of an antibody commendably overcomes the drawbacks of prodrug targeting. This conjugating model may provide a new general strategy for ADC design that is based on amino-containing cytotoxins.

The produced ADC showed significant stability improvement, which was a basic requirement of the linkers for MMAE-based ADCs. The plasma half-life of the prepared diisopropyl-substituted silyl ether-based linker was longer than 7 days, which was greatly improved compared with the half-life of the current hydrazone and carbonate type linkers (t_1/2_ ~ 2–3 days or 1 day, respectively) [2,17,52,53,54]. We speculated that the ideal steric hindrance of the diisopropyl-substituted silyl ether structure better guarantees the stability of the ADC in neutral environments. Further studies showed that this ADC can quickly and effectively release the payload in weakly acidic buffer, in which the rate and extent of drug release show dependence on acidity. The above features of this linker system fully meet the two basic requirements of the MMAE-based ADCs for linkers.

Further in vitro cell viability studies demonstrated that the ADC showed potent efficacy and significant antigen selectivity. Endocytic studies also showed that the ADC can be effectively internalized into target cells and eventually enter the lysosomes, where the weakly acidic environment (pH ~ 4.5) will ultimately ensure the effective drug releases of the ADC. In addition, the microtubule polymerization inhibition test showed that the ADC’s potential metabolite, Cys-6, can also produce microtubule polymerization inhibition that is similar to MMAE, meaning that even if free MMAE is not completely released, the metabolites can also work synergistically. This may be another reason for the increased cytotoxicity of this ADC. Furthermore, cell cycle arrest assay indicated that the ADC can suppress the cell cycle to the G_2_/M phase, which is most likely caused by inhibition of microtubule polymerization, further elaborates the anti-tumor mechanism. The properties exhibited by this silyl ether-based acid-cleavable ADC provide important support and inspiration for our future research.

In the pharmacodynamic study of xenograft models in vivo, this ADC not only exhibited superior tumor inhibition than naked antibody mil40 (*p* = 0.0054), but also showed significant dose-effect dependencies. Compared to the usual administration frequency of the acid-cleavable ADCs of about 2 times per week [17,52,55], the tumor growth inhibition rate can be as high as 93.6% for this ADC with weekly dosing treatments (5 mg/kg). Moreover, there was no obvious adverse reaction during the treatment of animals, as the body weight of all of the tested mice increased steadily. In addition, there were no significant differences in hematological parameters between ADC-treated mice compared with the control group (*p* > 0.05), further indicating a lower therapeutic toxicity. In vivo studies showed that this novel acid-cleavable ADC with MMAE as a potent payload initially maintains a balance between efficacy and toxicity, which was expected to make it a potential leader for the development of acid-cleavable MMAE-based ADCs.

## 4. Materials and Methods

### 4.1. Chemistry

Unless stated otherwise, all of the anhydrous solvents were commercially obtained and stored in sure-seal bottles under dry nitrogen. All of the other reagents and solvents were purchased at the highest grade available and used as received. Anti-HER2 antibody mil40 (a biosimilar of Herceptin^®^) was purchased from Hisun Pharmaceutical CO., Ltd. (Taizhou, Zhejiang, China). Monomethyl auristatin E (MMAE) was purchased from Concortis Biosystems (San Diego, CA, USA). Zenon™ pHrodo™ iFL red human IgG labeling reagent (Z25612), NucBlue™ Live ReadyProbes™ reagent (R37605), and CellLight™ Lysosomes-GFP, BacMam 2.0 (C10507) were purchased from Invitrogen (Carlsbad, CA, USA).

Thin layer chromatography was performed using pre-coated silica gel plates (Yantai Dexin Bio-Technology Co., Ltd., Yantai, China). Column chromatography was performed using silica gel (200–300 mesh; Yantai Chemical Industry Research Institute, Yantai, China). Proton nuclear magnetic resonance (^1^H-NMR) spectra were recorded on a JNM-ECA-400 400 MHz spectrometer (JEOL Ltd., Tokyo, Japan) using CDCl_3_ as solvent. Chemical shifts (δ) are expressed in ppm, with tetramethylsilane (TMS) functioning as the internal reference, where (δ) TMS = 0.00 ppm. The mass spectrometry (MS) systems used were the API 3000 triple-quadrupole mass spectrometer equipped with a Turbo Ion Spray electrospray ionization (ESI) source (AB Sciex, Concord, ON, Canada) for liquid chromatography-mass spectrometry (LC/MS) analysis and the Agilent G6230A mass spectrometer for accurate mass detection (Agilent, Santa Clara, CA, USA). Both of them were equipped with a standard ESI source.

### 4.2. Synthesis of the Silyl Ether-Based Linker-MMAE Conjugate

The synthetic route for the linker-MMAE conjugate (compound 6) was shown in Figure 2, with the original Mass and NMR spectrogram details of the above synthesized compounds provided in the Supporting Information, Appendix A.

#### 4.2.1. Synthesis of 1-(2-(2-hydroxyethoxy)ethyl)-1*H*-pyrrole-2,5-dione (2)

Compound 2 was synthesized using previously reported experimental protocols [56]. Maleimide (4.0 g, 41.2 mmol) was dissolved in ethyl acetate (160 mL), then cooled to 0 °C in ice water. *N*-methylmorpholine (3.7 g, 41 mmol) was then added slowly in a dropwise fashion after stirring for 10 min. A solution of methyl chloroformate (4.92 g, 40 mmol) in ethyl acetate (10 mL) was slowly added dropwise and the reaction continued for 1 h at room temperature. After the completion of the reaction, the insoluble salt was removed by filtration. The filtrate was concentrated to give a crude product methyl 2,5-dioxo-2,5-dihydro-1*H*-pyrrole-1-carboxylate (1), which was obtained as a red oily solid (6.55 g, 99% yield) after storage in low temperature. The product was unstable which was directly subjected to the next reaction without further purification.

2-(2-Aminoethoxy)ethan-1-ol (4.0 g, 41.2 mmol) was dissolved in a saturated sodium bicarbonate solution (120 mL) and cooled to 0 °C. After the desired temperature was reached, the above solution was transferred to compound 1 (6.55 g, crude product) and the reaction was continued at 0 °C for 30 min. The sample was slowly warmed to room temperature and continued to react for 1 h. After the completion of the reaction, the mixture was extracted with chloroform. The crude product was purified by column chromatography to obtain compound 2 as a white solid (4.37 g, 57% yield of two-step reaction). ^1^H-NMR (400 MHz, CDCl_3_): *δ* 6.73 (s, 2H), 3.75 (t, *J* = 5.1 Hz, 2H), 3.69 (t, *J* = 5.1 Hz, 2H), 3.65 (t, *J* = 5.3 Hz, 2H), 3.57 (t, *J* = 5.3 Hz, 2H), 2.27 (br, 1H). ESI *m*/*z* (M + H)^+^ calculated for C_8_H_12_NO_4_ 186.07; found 186.07. ESI *m*/*z* (M + Na)^+^ calculated for C_8_H_11_NNaO_4_ 208.06; found 208.05.

#### 4.2.2. Synthesis of 4-(((2-(2-(2,5-dioxo-2,5-dihydro-1*H*-pyrrol-1-yl)ethoxy)ethoxy)diisopropylsilyl)-oxy) benzaldehyde (3)

Compound 2 (0.5 g, 2.7 mmol) and triethylamine (0.55 g, 5.4 mmol) were dissolved in dichloromethane (20 mL) and the mixture was cooled to 0 °C. A solution of dichlorodiisopropylsilane (0.6 g, 2.7 mmol) in DCM (10 mL) was slowly added dropwise. After 40 min, p-hydroxybenzaldehyde (0.33 g, 2.7 mmol) in tetrahydrofuran (5 mL) was slowly added dropwise for an additional 1 h of reaction. After completion of the reaction, the solvent was concentrated to give a crude material, which was purified by column chromatography to obtain compound 3 as a white solid (0.86 g, 76% yield). ^1^H-NMR (400 MHz, CDCl_3_): *δ* 9.90 (s, 1H), 7.80 (dt, *J* = 8.4 Hz, 2H), 7.08 (dt, *J* = 8.4 Hz, 2H), 6.69 (s, 2H), 3.92 (t, *J* = 5.1 Hz, 2H), 3.72 (t, *J* = 5.1 Hz, 2H), 3.64 (t, *J* = 5.3 Hz, 2H), 3.58 (t, *J* = 5.3 Hz, 2H), 1.16 (m, 2H), 1.05 (m, 12H). ^13^C-NMR (100 MHz, CDCl_3_): *δ* 191.10, 170.73, 161.05, 134.23, 132.07, 130.58, 120.26, 71.84, 67.98, 62.99, 37.31, 17.20, 12.39. HRMS (ESI) *m*/*z* (M + H)^+^ calculated for C_21_H_30_NO_6_Si 420.1842; found 420.1838. HRMS (ESI) *m*/*z* (M + Na)^+^ calculated for C_21_H_29_NNaO_6_Si 442.1662; found 442.1655.

#### 4.2.3. Synthesis of 1-(2-(2-(((4-(hydroxymethyl)phenoxy)diisopropylsilyl)oxy)ethoxy)ethyl)-1*H*- pyrrole-2,5-dione (4)

Compound 3 (0.5 g, 1.19 mmol) was dissolved in dry THF (15 mL) and the mixture was cooled to -5 °C after dissolution. Sodium borohydride (18 mg, 0.48 mmol) was added to the reaction mixture and the mixture was reacted for 2 h and then filtered to remove the insoluble salt; the filtrate was concentrated under reduced pressure and evaporated to dryness to give a crude product, which was purified by column chromatography to obtain compound 4 as a colorless oil (0.1 g, 23% yield). ^1^H-NMR (400 MHz, CDCl_3_): *δ* 7.25 (t, 2H), 6.93 (m, 2H), 6.67 (s, 2H), 4.61 (s, 2H), 3.90 (t, *J* = 5.2 Hz, 2H), 3.69 (t, *J* = 5.0 Hz, 2H), 3.61 (t, *J* = 5.3 Hz, 2H), 3.56 (t, *J* = 5.2 Hz, 2H), 1.73 (br, 1H), 1.13 (m, 2H), 1.05 (m, 12H). ESI *m*/*z* (M + Na)^+^ calculated for C_21_H_31_NNaO_6_Si 444.5; found 444.4.

#### 4.2.4. Synthesis of 4-(((2-(2-(2,5-dioxo-2,5-dihydro-1*H*-pyrrol-1-yl)ethoxy)ethoxy)diisopropylsilyl)-oxy) benzyl(4-nitrophenyl) carbonate (5)

Compound 4 (253 mg, 0.6 mmol) was dissolved in dry DMF (15 mL) and bis(p-nitrophenyl) carbonate (360 mg, 1.2 mmol) and DIPEA (104 mg, 0.9 mmol) were added; the reaction was stirred at room temperature for 12 h. After the completion of the reaction, the solvent was concentrated under reduced pressure to remove the solvent. The obtained residue was subjected to diethyl ether dispersion and then filtered to obtain crude product, which was further purified by column chromatography to obtain compound 5 as a white solid powder (253 mg, 72% yield). ^1^H-NMR (400 MHz, CDCl_3_): *δ* 8.28 (dt, *J* = 9.0 Hz, 2H), 7.38 (dt, *J* = 9.2 Hz, 2H), 7.33 (dt, *J* = 8.4 Hz, 2H), 6.98 (dt, *J* = 8.7 Hz, 2H), 6.68 (s, 2H), 5.23 (s, 2H), 3.91 (t, *J* = 5.2 Hz, 2H), 3.72 (t, *J* = 5.6 Hz, 2H), 3.65 (t, *J* = 5.6 Hz, 2H), 3.58 (t, *J* = 5.2 Hz, 2H), 1.14 (m, 2H), 1.06 (dd, 12H). ESI *m*/*z* (M + NH_4_)^+^ calculated for C_28_H_38_N_3_O_10_Si 604.3; found 604.6. ESI *m*/*z* (M + Na)^+^ calculated for C_28_H_34_N_2_NaO_10_Si 609.2; found 609.5.

#### 4.2.5. Synthesis of the Linker-MMAE Conjugate (Compound 6)

Compound 5 (35.2 mg, 0.06 mmol) was dissolved in dry DMF (3 mL), then HOBt (8.28 mg, 0.06 mmol), MMAE (40 mg, 0.06 mmol), and DIPEA (10.68 μL) were added; the reaction was stirred at room temperature for 18 h. After the reaction, the solvent was concentrated under reduced pressure to remove the solvent. The obtained residue was further purified by column chromatography to afford compound 6 as a white solid powder (43.4 mg, 62% yield). ^1^H-NMR (400 MHz, CDCl_3_): *δ* 7.38-7.31 (m, 6H), 7.25-7.20 (t, *J* = 8.12 Hz, 2H), 7.13 (d, *J* = 8.12 Hz, 0.5H), 7.02 (d, *J* = 9.28 Hz, 0.5H), 6.90 (d, *J* = 8.44 Hz, 2H), 6.68 (s, 2H), 6.55 (m, 1H), 5.15-5.03 (m, 2H), 4.95 (d, *J* = 2.8 Hz, 1H), 4.69 (m, 2H), 4.28-4.05 (m, 5H), 3.91 (t, *J* = 5.05 Hz, 2H), 3.72 (t, *J* = 5.50 Hz, 2H), 3.64 (t, *J* = 5.50 Hz, 2H), 3.57 (t, *J* = 5.05 Hz, 2H), 3.52-3.50 (m, 2H), 3.40 (m, 4H), 3.30 (s, 3H), 3.11 (m, 1H), 3.01 (m, 3H), 2.87 (m, 3H), 2.49-2.32 (m, 3H), 2.21 (m, 1H), 2.04 (m, 3H), 1.85 (m, 2H), 1.33-1.24 (m, 5H), 1.04 (m, 19H), 0.98-0.77 (m, 26H). ^13^C-NMR (100 MHz, CDCl_3_): *δ* 174.78, 170.77, 170.54, 169.98, 169.70, 157.58, 155.18, 141.21, 134.21, 130.07, 129.55, 128.05, 127.39, 126.44, 119.65, 82.04, 78.61, 75.84, 71.94, 67.96, 67.50, 64.95, 62.80, 61.08, 60.19, 58.10, 57.25, 54.21, 51.64, 47.96, 45.10, 37.73, 37.36, 33.33, 32.03, 31.00, 29.81, 26.21, 25.81, 25.10, 25.04, 19.41, 18.61, 17.29, 16.04, 14.54, 14.00, 12.38, 10.98. HRMS (ESI) *m*/*z* (M + H)^+^ calculated for C_61_H_97_N_6_O_14_Si 1165.6832; found 1165.6828. HRMS (ESI) *m*/*z* (M + Na)^+^ calculated for C_61_H_96_N_6_NaO_14_Si 1187.6651; found 1187.6645.

### 4.3. Preparation of Reagents and Antibody-Drug Conjugates

Reagents and antibody-drug conjugates were prepared with methods as described [38]. Briefly, humanized anti-HER2 antibodies mil40 (10 mg/mL) in L-histidine buffer (20 mM, pH ~ 7.5), were treated with TCEP (3.0 equivalents) at 25 °C for 90 min. To the reduced mAb was added the maleimide drug derivative 6 (2 equivalents/SH group) in dimethylacetamide (DMAC) (8% *v*/*v*). After 120 min, the reactions were quenched with excess NAC. The mixture was adjusted to weak acidity (pH ~ 5.5) with diluted AcOH (0.3 mol/L) and placed on ice for 30 min before buffer exchange by elution through Sephadex G25, and concentrated by centrifugal ultrafiltration to obtain target ADC (mil40-6). The conjugates’ concentrations were determined by UV absorption at 280 nm, and were filtered through a 0.2-μm filter under sterile conditions and stored at −80 °C before use for analysis and testing.

### 4.4. The Stability Assays of Linker and ADC in Plasma

Human plasma (LOT#: BRH1343165) was purchased from BioreclamationIVT (Westbury, NY, USA). NAC (900 μL, 0.41 mg/mL in PBS, pH 7.4) was added to linker-MMAE 6 (100 μL, 70.9 μg/mL in DMSO) and incubated in a water bath at 25 °C for 10 min until complete conversion to NAC-6 [38]. The reaction mixture was mixed into an equal volume of human plasma and incubated in a sterile incubator at 37 °C. Aliquots (50 μL) were collected at subsequent time points (0, 3, 6, 12, 24, 36, 48, 72, 96, 120, 144, and 168 h) and quenched with cold acetonitrile (150 μL) before frozen at −80 °C. After sampling was completed, all of the samples were melted at room temperature, centrifuged to remove protein, and analyzed by LC/MS. Results were based on the area under curve (AUC) of the off-target MMAE at each time point, with the total amount of MMAE contained in the substrate defined as 100%. Analogously, ADC (mil40-6) in PBS (1.8 mg/mL, pH 7.4) was mixed into an equal volume of human plasma and incubated in a sterile incubator at 37 °C. Similarly, aliquots (50 μL) were collected at subsequent time points (0, 3, 6, 12, 24, 36, 48, 72, 96, 120, 144, and 168 h) and quenched with acetonitrile (150 μL) before being frozen at −80 °C. After sampling was completed, the off-target MMAE were detected using the methods described above.

### 4.5. Effect of Acidity Difference on the Drug Release Process

The drug release process was characterized using DAR values over time. ADC (mil40-6) in PBS (2.0 mg/mL, 9 mL, pH 7.4) was divided into 3 parts; then, diluted acetic acid solution (0.3 mmol/L) was added to two of them to adjust the pH value to 4.5 and 5.5. The resulting three solutions were placed in an incubator at 37 °C for incubation; aliquots (100 μL) were collected at subsequent time points (0, 3 h, 6 h, 12 h, 24 h, 2 d, 3 d, 4 d, and 7 d) and frozen at −80 °C. After sampling was completed, the DAR of each sample was determined using hydrophobic interaction chromatography high-performance liquid chromatography.

### 4.6. Evaluation of ADC for Cancer Cell Killing in Vitro

The human HER2-positive cell lines (BT-474, NCI-N87, and MDA-MB-453), the HER2-normal level cell line (MCF-7), and the HER2-negative cell line (MDA-MB-231) were all purchased from American Type Culture Collection (ATCC, Manassas, VA, USA). All of the cell lines were cultured in medium supplemented with 10% fetal bovine serum (FBS) at 37 °C in a humidified atmosphere containing 5% CO_2_ in air; cells were subcultured three times per week at a ratio of 1:4 in medium with 10% FBS and 1% penicillin/streptomycin solution. Cells (3.3 × 10^4^ cells/mL) were added to each well of a 384-well plate and incubated at 37 °C overnight, after which 10 μL compound aliquots were added to the assay plate. The plate was incubated for 3 days at 37 °C, 5% CO_2_, and 95% humidity. Then, the plates were incubated at room temperature for about 10 min; 40 μL Cell Titer Glo^®^ (CTG) reagent was added to each well and the plates were incubated for 30 min at room temperature. Luminescence was detected using the EnSpire Plate Reader (PerkinElmer, Waltham, MA, USA). All of the experiments were conducted independently and in triplicate. The effects of each agent on the proliferation of cancer cell lines were expressed as the % cell growth inhibition using the following formula: Remaining activity (%) = (S–M)/(V–M) × 100%, in which S is the readout of the test sample, V is the readout of the vehicle sample, and M is the readout of the well with compounds treatment. The 50% inhibiting concentration (IC_50_) was calculated by Prism5 for Windows (Graphpad Software Inc., La Jolla, CA, USA).

### 4.7. Endocytosis and Transport for the Silyl Ether-Based ADC

The lysosomal trafficking of the resulting ADC was examined with reference to a published method [46]. BT-474 cells (6 × 10^5^/well) were seeded in 24-well plates containing glass covers and lips and was cultured with Dulbecco’s Modified Eagle Medium (DMEM) plus 10% fetal bovine serum (FBS) at 37 °C overnight. Cells were further incubated with CellLight Lysosomes-green fluorescent protein (GFP, C10507, Invitrogen, Carlsbad, CA, USA) for 30 min on ice or 16 h at 37 °C; then, the lysosomal membrane protein Lamp-1 was labeled with GFP. The antibody was labeled as red fluorescence (Invitrogen, Z25612; Zenon^®^ Labeling Technology, Carlsbad, CA, USA) with a final concentration of 6 μg/mL and was incubated with the labeled BT-474 cells for 16 h at 37 °C. Nucblue (Invitrogen, R37605) was added to the medium and incubated for another 15–30 min at 37 ° C; then, the cells were washed with cold PBS twice. Fluorescence images were taken using a Nikon A1 confocal microscope (Nikon Corp., Tokyo, Japan).

### 4.8. Microtubule Polymerization Assay

The Tubulin Polymerization Assay Kit (Cat. # BK011P, Cytoskeleton, Inc., Denver, CO, USA) was used according to the manufacturer’s instructions for the fluorescence-based test. Before this test, the silyl ether-based linker-MMAE conjugate (compound 6, 0.3 mmol/L in DMSO, 100 μL) was added to the L-cysteine solution (0.3 mmol/L in H_2_O, 900 μL). The reaction mixture was incubated at room temperature for 10 min, after which HPLC analysis revealed complete conversion to Cys-linker-MMAE (Cys-6), as previously reported [57]. MMAE and the Cys-linker-MMAE conjugate were used at 3.0 μM and 0.3 μM, respectively.

### 4.9. Cell Cycle Arrest in HER2-Positive Cancer Cells

Effects of the novel silyl ether-based ADC (10 ng/mL, 50 ng/mL, and 100 ng/mL; the concentration of the ADC was calculated as the concentration of the antibody) on the cell cycle were examined using the HER2-positive breast cancer cell line BT474, and using vehicle as a comparator. Cells were cultured with RPMI 1640 medium containing 10% FBS in a humidified atmosphere, each plate was seeded with 2 × 10^5^ cells for each group, and then normal cultured overnight at 37 °C, 5% CO_2_, and 95% humidity before adding the drug. The ADC was diluted with medium and added into the corresponding wells, and the medium was renewed every 3 days during the 6 days’ treatments. The Cell Cycle and Apoptosis Analysis Kit (C1052, Beyotime, Shanghai, China) was applied for PI labeling. The entire procedure was performed as outlined in the manufacture’s of manual. Results were analyzed with NovoCyte (ACEA Biosciences, San Diego, CA, USA).

### 4.10. In Vivo Antitumor Activity in Human Gastric Xenograft Tumors

Female BALB/c nude mice (6–8 weeks old) were purchased from Beijing Ankaiyibo Biological Technology Co. Ltd. (Beijing, China) and were acclimated for 1 week before the experiment. All of the animal experiments were performed by following the protocol approved by the Institutional Animal Care and Use Committee at Pharmaron Co., Ltd. (ON-CELL-XEM-06012017, 1 June 2017 of approval) To develop the human tumor xenografts, female nude mice, with an average weight of approximately 22 g, were inoculated subcutaneously with 5 × 10^6^ NCI-N87 gastric cancer cells in 0.2 mL of serum-free RPMI1640 medium for tumor development, one tumor per mouse. When tumors reached ~ 120 mm^3^, mice were randomly divided into treatments and vehicle control groups (*n* = 6/group). The mice were given ADC (mil40-6), antibody (mil40), and vehicle (physiological saline) on days 0, 7, 14, and 21. For this subcutaneous xenograft model, tumor volume and body weight were monitored 2–3 times per week from day 0 by electronic calipers during the treatment period, which was calculated using the formula: TV = a × b^2^/2, where “a” and “b” are long and short diameters of a tumor, respectively. When the subcutaneous xenografts were over 2000 mm^3^, or at the end of the experiments, animals were sacrificed according to institutional guidelines.

### 4.11. Hematology Analysis during Treatment

One week after the end of administration of the nude mice NCI-N87 xenograft model (day 28), the animals were subjected to blood collection (100–180 μL) for whole blood analysis. For some smaller blood samples, the volume was diluted twice and the whole blood count was performed, with the number of cells counted and analyzed based on the corrected dilution factor. The measured blood cell parameters mainly include white blood cells (WBC), red blood cells (RBC), hemoglobin (HGB), platelets (PLT), neutrophil (Neut), lymphocytes (Lymph), monocytes (Momo), eosinophils (Eos), and large unstained cells (LUC). In this experiment, the vehicle group (saline) was selected as the control group. The final results were expressed as mean ± SD. The data was analyzed based on the value with a corrected dilution factor. For comparison of hematological indicators, unpaired two-tailed *t* tests for multiple comparisons were used. The level of significance was set at *p* < 0.05. Statistical analyses were performed using Prism5 for Windows.

## 5. Conclusions

Acid-cleavable linkers were first used in clinical ADCs, but their structural category is still limited, and its stability is usually insufficient. The application of novel acid-cleavable linker technology to produce stable ADCs allowed us to assess the critical impact of drug-linker design upon stability and in vivo efficacy. In this study, we generated a novel, silyl ether-based acid-cleavable ADC with MMAE as the potent payload. By introducing PHB, the ADC skillfully achieves efficient combination of amino-conjugated MMAE with acid-triggered silyl ether groups. The resulting ADC exhibits improved stability, effective release of payload, appropriate efficacy in vitro and in vivo, and controlled therapeutic toxicity. This strategy may deepen the cognition of acid-cleavable linker and provide a new conjugating model for MMAE. As an addition to the existing acid-cleavable linkers, this linker system is expected to provide more options for the development of ADCs that load high-efficiency cytotoxins.

## Figures and Tables

**Figure 1 cancers-11-00957-f001:**
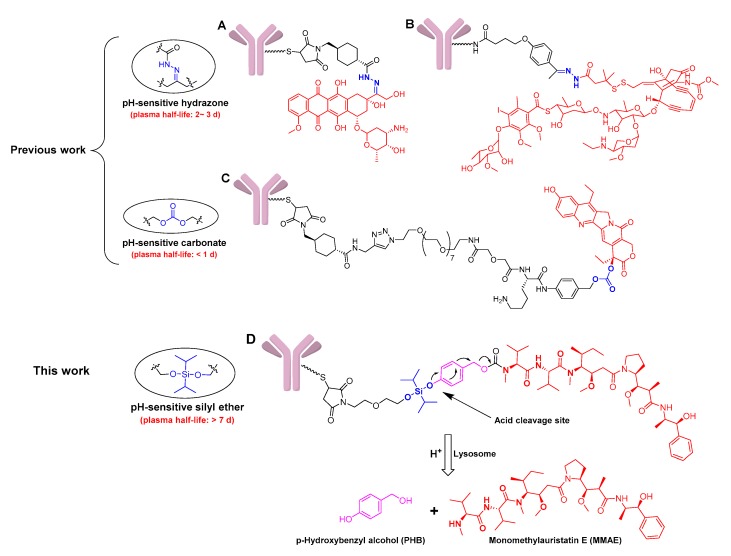
Structures of the chemically labile antibody-drug conjugates (ADCs) with acid-cleavable linkers, and the drug release mechanism of ADCs with monomethyl auristatin E (MMAE) as the payload. (**A**) The doxorubicin-based ADC with the pH-sensitive hydrazone linker; (**B**) the calicheamicin-based ADC with the pH-sensitive hydrazone linker; (**C**) the SN-38-based ADC with the pH-sensitive carbonate linker; (**D**) the designed MMAE-based ADC with the pH-sensitive silyl ether linker, and its drug release mechanism.

**Figure 2 cancers-11-00957-f002:**
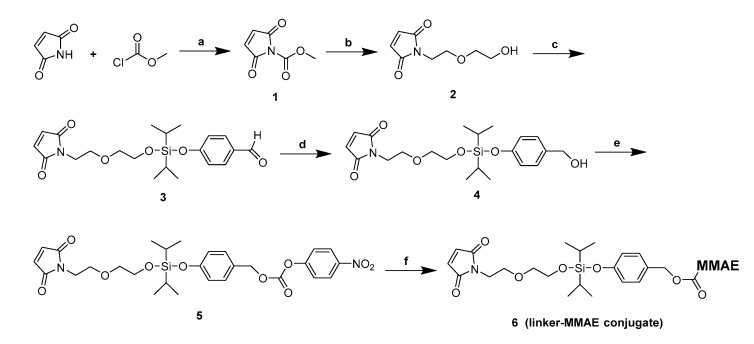
General synthetic route for the linker-MMAE conjugate (compound 6). Reagents and conditions: (**a**) NMM, EA, 0 °C, 2 h, 99%; (**b**) 2-(2-aminoethoxy)ethan-1-ol, NaHCO_3_, H_2_O, 0 °C ~ room temperature (rt), 1 h, 57%; (**c**) dichlorodiisopropylsilane, 4-hydroxybenzaldehyde, TEA, DCM, 0 °C ~ rt, 2 h, 76%; (**d**) NaBH_4_, THF, −5 °C–0 °C, 3 h, 23%; (**e**) Bis(4-Nitrophenyl) Carbonate, DIPEA, DMF, rt, 5 h, 72%; (**f**) MMAE, HOBt, DIPEA, DMF, rt, overnight, 62%.

**Figure 3 cancers-11-00957-f003:**
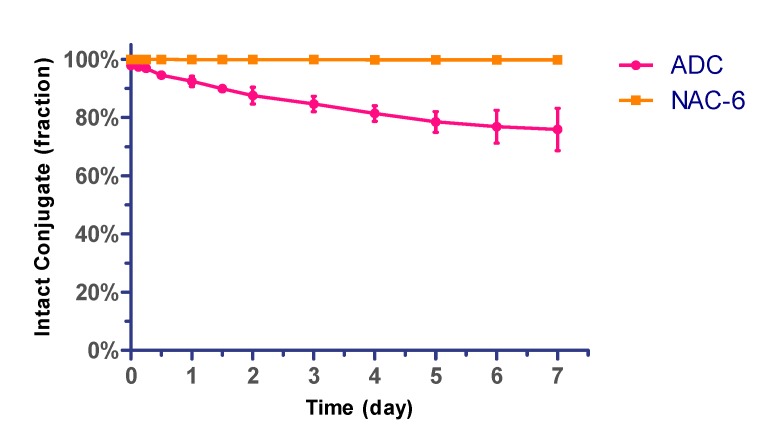
Stability assays of silyl ether-based ADC (mil40-6) and its payload conjugate (NAC-6) in human plasma. ADC and NAC-6 were incubated at 37 °C for 7 days. Error bars represent standard deviation from three independent experiments (triplicates). Results are shown as the mean ± SD. NAC: *N*-acetyl-*L*-cysteine.

**Figure 4 cancers-11-00957-f004:**
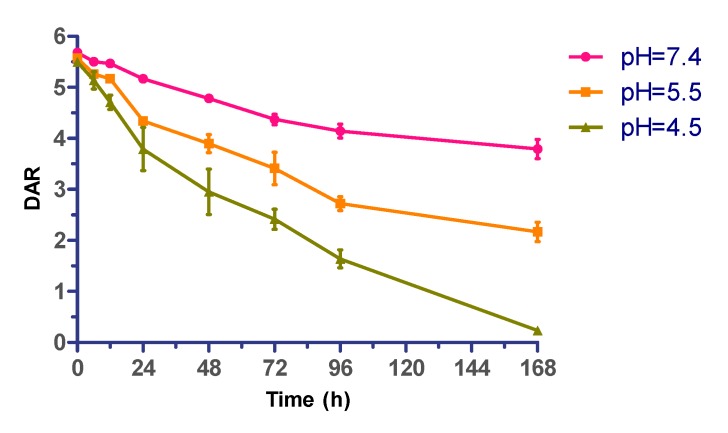
pH-dependent drug release of acid-sensitive ADC (mil40-6). Typical time course of MMAE cleaved from ADC in PBS that pH 7.4, pH 5.5, and pH 4.5. Error bars represent standard deviation from three independent experiments (triplicates). Data are shown as mean ± SD. DAR: drug/antibody ratio.

**Figure 5 cancers-11-00957-f005:**
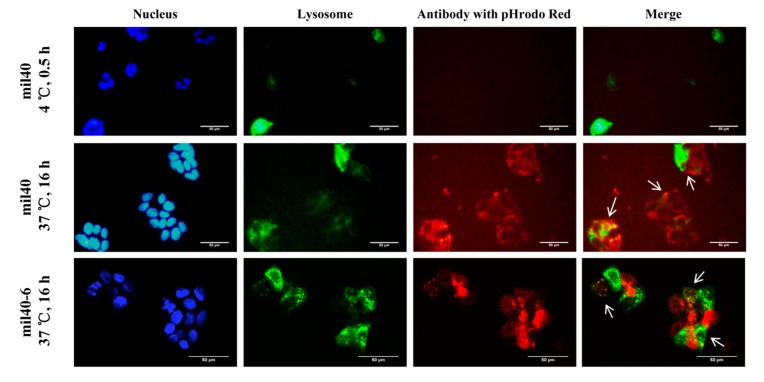
Receptor-mediated internalization of ADC mil40-6 by HER2^+^ breast cancer cell line BT-474. Red fluorescence-labeled mil40-6 (1 μg/mL) was incubated with GFP-labeled BT-474 cells at 37 °C for 16 h while mil40-6 was endocytosed into the cell and located in lysosomes. The cell nuclei were stained with DAPI (blue), the lysosomes were labeled with a lysosomes-GFP (green), and the antobody were labeled with pHrodo Red (red). GFP: green fluorescent protein; DAPI: 4′,6-diamidino-2-phenylindole. Scale bar = 50 μm.

**Figure 6 cancers-11-00957-f006:**
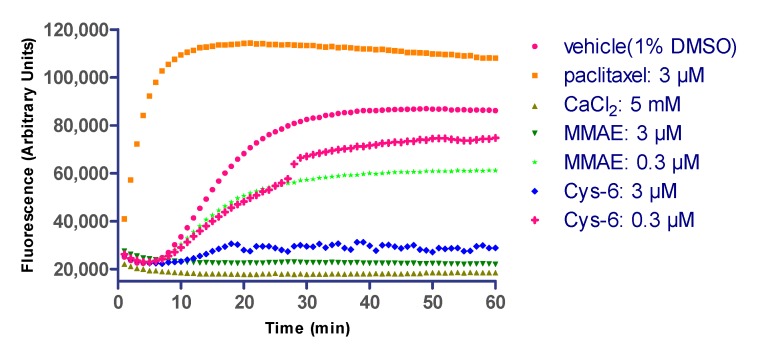
The inhibition tests of microtubule polymerization. The cytotoxin MMAE and the Cys-linker-MMAE (Cys-6) were tested in a microtubule polymerization assay. Free MMAE was included as a positive control. Paclitaxel and CaCl_2_ were included as a promoter and inhibitor, respectively, of microtubule polymerization. The buffer control indicates the polymerization rate of untreated microtubules.

**Figure 7 cancers-11-00957-f007:**
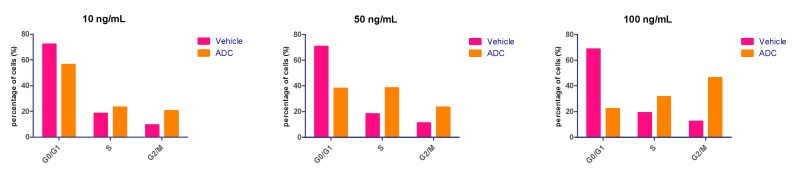
The effects of the ADC (mil40-6) on cell cycle arrest. The BT-474 cells cycle arrest results analyzed by flow cytometry (FCM). ADC was tested in three doses of 10 ng/mL, 50 ng/mL, and 100 ng/mL. The concentration of the ADC is calculated as the concentration of the antibody.

**Figure 8 cancers-11-00957-f008:**
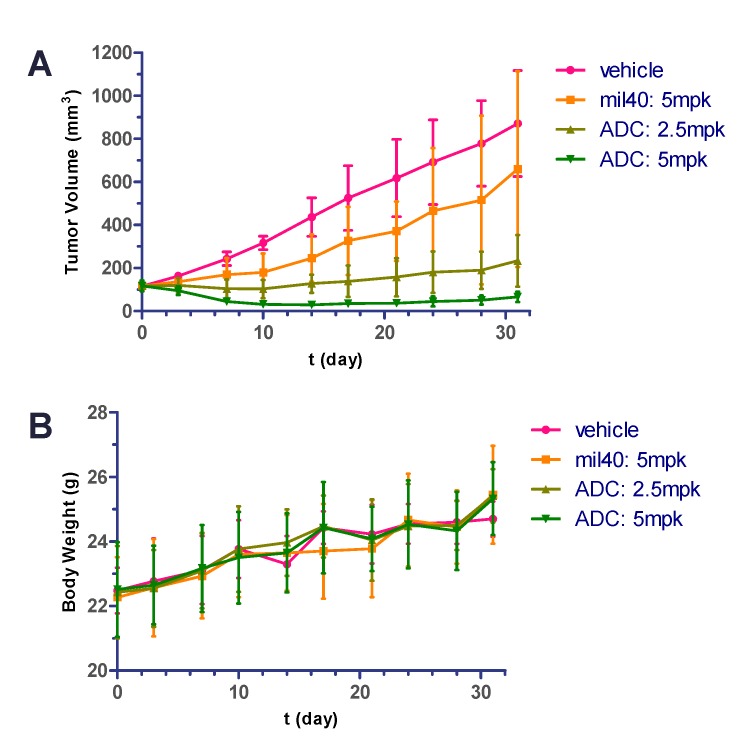
Xenograft studies of the ADC (mil40-6). Nude mice were implanted subcutaneously with NCI-N87 cells. When the tumors reached ~ 120 mm^3^, the animals were given vehicle, mil40, and ADC on: Days 0, 7, 14, and 21. Results are shown as mean ± SD, *n* = 6/group. (**A**) The tumor volume of the test animals during the treatment and observation period; (**B**) changes in body weight of the mice during the observation period. To compare the difference in activity between antibody and ADC, unpaired two-tailed *t* tests for multiple comparisons were used, and statistical analyses were performed using Prism5 for Windows.

**Figure 9 cancers-11-00957-f009:**
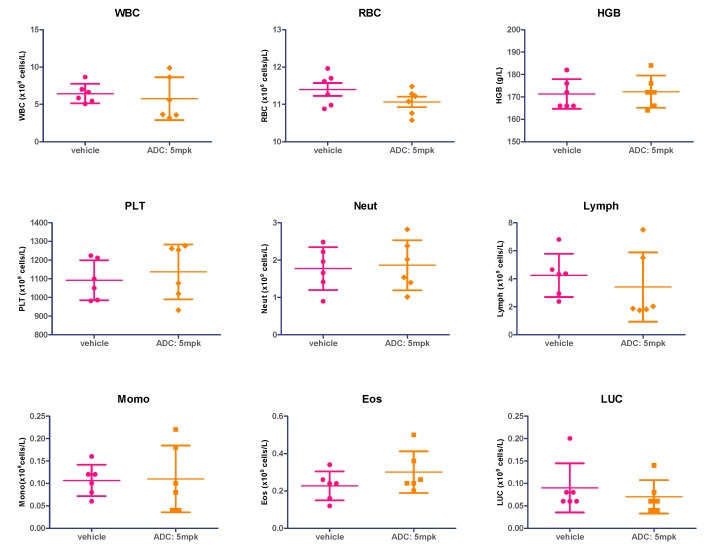
Hematological analysis of acid-sensitive ADC (mil40-6). WBC: white blood cells; RBC: red blood cells; HGB: hemoglobin; PLT: platelets; Neuts: neutrophils; Lymph: lymphocytes; Mono: monocyte; Eos: eosinophils; LUC: large unstained cell. Results are shown as mean ± SD, *n* = 6/group.

**Table 1 cancers-11-00957-t001:** Cytotoxicity of ADC, antibody and cytotoxin in cancer cell lines in vitro.

Cell Lines	HER2 Status	ADC		mil40		MMAE	
		IC_50_ (nM)	Max. Inhibition	IC_50_ (nM)	Max. Inhibition	IC_50_ (nM)	Max. Inhibition
BT-474	HER2^+++^	0.170	50.50%	0.323	61.25%	0.400	40.65%
NCI-N87	HER2^+++^	0.028	93.54%	1.157	45.80%	0.369	68.22%
MDA-MB-453	HER2^++^	0.101	87.30%	0.315	34.80%	0.295	83.79%
MCF-7	HER2^−^	7.742	47.8%	>1000	-	0.688	77.64%
MDA-MB-231	HER2^−^	>1000	-	>1000	-	1.156	53.4%

Note: the concentration of the ADC was calculated as the concentration of the antibody.

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
