# Peer review of "Novel Silyl Ether-Based Acid-Cleavable Antibody-MMAE Conjugates with Appropriate Stability and Efficacy"

_cancers, 2019, doi:10.3390/cancers11070957_

Round 1
Reviewer 1 Report
This is a very interesting work, with a novel approach to generate a new linkers technology for ADC development. Along with the development of monoclonal antibodies (mAbs) and cytotoxic drugs, the design of the linker is of essential importance, because it impacts the efficacy and tolerability of ADCs. Therefore, the relevance of the present research for drug development. In this study the authors used HER-2, a well-known target, and MMAE widely used as ADC payload, to probe the efficacy of the newly synthesised linker. Their data demonstrates the new linker preferentially releases the drug at acidic PH, the new ADC molecule called mil40-6 (DAR~5.5) is sufficiently stable hence it would be deemed to have stability during systemic circulation; and the conjugation technique doesn’t affect mil40-6 cytotoxic activity conferred by the payload either in vitro or in vivo. Additionally, no adverse effects were obtained when mil40-6 was evaluated in a murine model of Her2+ gastric cancer. Although this study will have benefited form direct side by side comparison of mil40-6 with a conventional Her2 ADC molecule such as Her2-vcE, the evidence presented support the data and claims made in the manuscript.
Minor
The authors provide evidence that mil40-6 internalises inside tumour cells, suggesting this ADC accumulates in the lysosomal compartment where it colocalizes with LAMP-1. However, this reviewer would like to see the quantitative evaluation of colocalization, which is needed to demonstrate that the overlap of two fluorophores is not random. The methods most substantially used are Pearson´s correlation coefficient (PCC) or Manders colocalization coefficient (MCC), and both methodologies can be used to asses colocalization (Costes et al. 2004; Dunn et al. 2011).
Methodology, please provide the source of human plasma use for the stability study.
It is very important that the final version of the paper is edited by a native English speaker - there are some grammatical issues to address throughout.
Avoid claims of novelty, such as "for the first time," even when qualified by clauses such as "to our knowledge.”
Figure 1. it is very difficult to see the pH sensitive chemical structures.
Line 145, what is the meaning of Her2- normal level cell lines? Better to classified as Her2 positive and negative cell lines.
Legend of figure 4, please described green indicates LAMP1 expression.
Line 239 simple not simplex.
Author Response
Comment 1:
The authors provide evidence that mil40-6 internalises inside tumour cells, suggesting this ADC accumulates in the lysosomal compartment where it colocalizes with LAMP-1. However, this reviewer would like to see the quantitative evaluation of colocalization, which is needed to demonstrate that the overlap of two fluorophores is not random. The methods most substantially used are Pearson´s correlation coefficient (PCC) or Manders colocalization coefficient (MCC), and both methodologies can be used to asses colocalization (Costes et al. 2004; Dunn et al. 2011).
Response:
Thank you for your valuable and thoughtful comments. At present, the endocytosis studies of ADCs usually only perform qualitative analysis. In our manuscript, we have only conducted a qualitative analysis of the lysosomal pathway of antibody and ADC, which may not be sufficient. According to your suggestion, we replaced the image to make the results clearer. Furthermore, we have calculated the correlation of the overlap of two fluorophores (green and red), and supplemented the Pearson correlation coefficient to the relevant description of the manuscript (section 2.6; page 6).
Comment 2:
Methodology, please provide the source of human plasma use for the stability study.
Response:
Thank you for your kind reminder. We have already supplemented the source information of human plasma in the manuscript (section 4.4; page 13): “the human plasma was purchased from BioreclamationIVT (LOT#: BRH1343165)”.
Comment 3:
It is very important that the final version of the paper is edited by a native English speaker - there are some grammatical issues to address throughout.
Response:
We sincerely apologize for the grammatical issues in our manuscript. Further, we have tried our best to make a comprehensive inspection and adjustment of the manuscript according to your suggestion.
Comment 4:
Avoid claims of novelty, such as "for the first time", even when qualified by clauses such as "to our knowledge.”
Response:
We have modified the manuscript according to your suggestion, for example, deleting the description of “for the first time”.
Comment 5:
Figure 1. It is very difficult to see the pH sensitive chemical structures.
Response:
We have adjusted the pH-sensitive chemical structures in Figure 1 to make it clearer (page 2~3).
Comment 6:
Line 145, what is the meaning of Her2-normal level cell lines? Better to classified as Her2 positive and negative cell lines.
Response:
Thank you for your valuable comment. Since the expression level of HER2 on normal cells is also negative, we finally classify MCF-7 cell into HER2-negative tumor cell lines (section 2.5; page 5) according to other related references (Mol Cancer Ther, 2013, 12(9):1816-1828; Eur J Pharm Sci, 2016, 93:274-286; Breast Cancer Res Treat, 2015,153(1):123-133.).
Comment 7:
Legend of figure 4, please described green indicates LAMP-1 expression.
Response:
We have supplemented the relevant expression of “green indicates LAMP-1” in the legend of Figure 4 (page 6).
Comment 8:
Line 239 simple not simplex.
Response:
We have replaced “simplex” with “simple” according to your suggestion (section 3; page 9).
We sincerely thank you for your warm work and your suggestions, which are very valuable for us to further improve our manuscript. We will do our best to improve the manuscript and made some changes, and hope that the correction will meet with your approval.

Reviewer 2 Report
Wang et al developed and employed a novel silyl ether-based acid-cleavable linker to conjugate MMAE to anti-Her2 antibody (ADC). The authors analyzed the stability of this linker/ADC in plasma and acidic environments and observed good stability. The authors demonstrate that the ADC has potent in vitro anti-tumor activity against Her(high) and Her(low) tumor cell lines, but not against Her(negative) and non-tumorigenic cell lines. Finally, the authors demonstrate the in vivo efficacy and safety of ADC.
In general, the study was performed well (although, the inclusion of a ADC with a different commonly-used linker would have improved the impact). Importantly, few concerns have to be addressed before the publication of this manuscript in this journal:
1) Line 40-41 should be modified to describe the motivation behind acid-cleavable linkers - difference in pH in the endolysosomal compartments and extracellular space. And original reference should be included.
2) Different Y-axis (intact conjugate versus DAR) is used for Figure 2 and 3. If possible, the authors should use the same variable to study the effects of serum and pH. If not, the authors should clarify the choice.
3) The authors should discuss the reasons for the following: i) maximum ADC-induced inhibition of only approx. 50% for BT474 and ii) maximum inhibition of BT474 was greater with mil40 versus ADC.
4) In line 155, the word 'cytoplasm' should be replaced by 'acidic environment of endosomal or lysosomal compartments' or similar.
5) In Figure 4, the regions highly positive for ADC are negative for lysosome staining - the authors should clarify this.
6) In Figure 7, statistics (one-way or two-way ANOVA) should be included.
7) In Figure 8, Student's t-test should be used for each parameter.
8) The authors should discuss the drawbacks of Val-Cit-based cathepsin B-cleavable linker.
Author Response
Comment 1:
Line 40-41 should be modified to describe the motivation behind acid-cleavable linkers - difference in pH in the endolysosomal compartments and extracellular space. And original reference should be included.
Response:
Thank you for your valuable advice. We have added the descriptions of pH value about tumor tissue and plasma, and added related references at the relevant locations (section 1; page 1).
Comment 2:
Different Y-axis (intact conjugate versus DAR) is used for Figure 2 and 3. If possible, the authors should use the same variable to study the effects of serum and pH. If not, the authors should clarify the choice.
Response:
We are very grateful to your comment. If the substances being analyzed are the same or similar, it would be better to use the same variables to study the effects of serum and pH. As shown in Figure 2, we evaluated the in vitro plasma stability of NAC-linker-MMAE conjugate and the related ADC, so their co-released products (MMAE) were tested. However, we showed the acid-dependent drug release process of ADC in Figure 3, since the average drug/antibody ratio (DAR) at different time points can more accurately reflect the immediate characteristics of the ADC, we used DAR to reduce the error as our previous methods (Int. J. Mol. Sci., 2017, 18, 1860).
Comment 3:
The authors should discuss the reasons for the following: i) maximum ADC-induced inhibition of only approx. 50% for BT474 and ii) maximum inhibition of BT474 was greater with mil40 versus ADC.
Response:
Thank you for your thoughtful comments. i) in this study, since the breast tumor cell line BT-474 grew at a slower rate, the number of passages was relatively less during the limited incubation time (72 h) after administration, resulting in a maximum inhibition rate of only about 50%. ii) the naked antibody mil40 is a biosimilar of trastuzumab (Herceptin), we have demonstrated that these two antibodies have similar antitumor activity on HER2-positive tumor cell lines such as BT-474, SK-BR-3, NCI-N87, and SK-OV-3 (Int. J. Mol. Sci., 2017, 18, 1860.); since BT-474 cells are very sensitive to mil40 itself, the efficacy of the antibody masks the performance of the ADC to some extent. As a result, both of them show high antitumor activity (IC50 ~ 10-10 nM) and a similar maximum inhibition rate.
Comment 4:
In line 155, the word 'cytoplasm' should be replaced by 'acidic environment of endosomal or lysosomal compartments' or similar.
Response:
Lysosomal is the main metabolic site of ADCs (Bioconjugate Chem, 2017, 28, 1102-1114.), according to your good suggestion, we have replaced “cytoplasm” with “lysosomal compartments” (section 2.6; page 5).
Comment 5:
In Figure 4, the regions highly positive for ADC are negative for lysosome staining - the authors should clarify this.
Response:
Thank you for your thoughtful comment. Labeled antibodies co-localize with lysosomes, relying on the sufficient concentration of antibody in lysosomes. As shown in Figure 4, we can see the yellow color produced by the merge of red (antibody) and green (LAMP-1), demonstrating that antibody can enter the lysosomes; however, the intensity of the two fluorescences we observed may be locally non-uniform. Based on your comment, we replaced some images to make the results clearer (page 6). Moreover, we performed a quantitative analysis of the overlap of the two fluorescences and supplemented the Pearson correlation coefficient in the manuscript (section 2.6; page 6).
Comment 6:
In Figure 7, statistics (one-way or two-way ANOVA) should be included.
Response:
According to your kind reminder, we supplemented the relevant expression of statistics in the legend of Figure 7 (page 8).
Comment 7:
In Figure 8, Student's t-test should be used for each parameter.
Response:
We excluded the non-representative hematology parameters (section 2.10; page 9), and added the conclusion of an unpaired two-tailed t-test of common hematological parameters at the corresponding position in the manuscript (section 2.10; page 8).
Comment 8:
The authors should discuss the drawbacks of Val-Cit-based cathepsin B-cleavable linker.
Response:
Thanks again for your careful work. In the introduction, we have briefly described the drawbacks of Val-Cit-based cathepsin B-cleavable linkers, however, it is undeniable that it still plays an important role in current ADCs research. As the Val-Cit-based linkers are still not perfect, researchers need to constantly develop new linkers to further advance the development of ADCs.

Reviewer 3 Report
The manuscript by Zhong, Zhou, and coworkers describes the first example of an antibody drug conjugate of monomethyl auristatin E which employs a dialkyl silyl ether as the acid-labile dissociable component, the hydrolysis of which triggers release of a 4-hydroxybenzyl ether of the free drug, and thereafter the free drug. The work is novel and appears useful in that the conjugates perform as desired, demonstrating good stability in neutral media while liberating the protected component at pH 7.4. The MMAE conjugate is shown to maintain useful levels of efficacy and acceptable levels of toxicity.
There are several issues that need to be addressed:
1. When a compound has been previously generated, the preparation should reference that work. In this case, the hydroxyethyloxyethyl amide of maleimide has been prepared multiple times (first? By Miyadera, J. Med. Chem. 1971, 873) and the method employed by the authors may stem from Weber, et al, Bioconj. Chem, 1990, 1 431). Why does the experimental not include appropriate statements and citations (“was prepared as per the report by Weber…”; “The spectra matched previous reports…”
2. For new molecules (structures 3,4, 5, and 6), no 13C NMR spectra is included in the experimental characterization or the SI. I don’t know the standard for this journal but such an assignment would be required in most organic and many medicinal-themed journals (e.g. J. Med. Chem.)
3. Scheme 1 should include yields, either within the graphic or else in the captions.
4. The previous work in the area (from the DiSimone group) is adequately cited. However, whereas the discussion does a nice job of explaining the similarities and differences between this work and the previous work (the previous work used silyl ethers as the direct precursors of active drug; this work uses the silyl ether as an acid-labile protecting element for a labile 4-hydroxybenzyl carbamate), the introduction (page 2, last paragraph) needs to be modified so that the reader will be aware that another group (DiSimone) looked at both trialkylsilyl and dialkyl silyl groups and, in the latter group, established the desirability of the diisopropylsilyl ethers employed here.
Author Response
Comment 1:
When a compound has been previously generated, the preparation should reference that work. In this case, the hydroxyethyloxyethyl amide of maleimide has been prepared multiple times (first? By Miyadera, J. Med. Chem. 1971, 873) and the method employed by the authors may stem from Weber, et al, Bioconj. Chem, 1990, 1 431). Why does the experimental not include appropriate statements and citations (“was prepared as per the report by Weber…”; “The spectra matched previous reports…”
Response:
Thank you for your valuable comment. We have made adjustments to the description of the synthetic method about compound 2 and added relevant references (section 4.2.1; page 11).
Comment 2:
For new molecules (structures 3,4, 5, and 6), no 13C NMR spectra is included in the experimental characterization or the SI. I don’t know the standard for this journal but such an assignment would be required in most organic and many medicinal-themed journals (e.g. J. Med. Chem.)
Response:
We agree with your rigorous attitude and thank you for your kind advice. J. Med. Chem. is a professional journal in the field of chemistry, which has strict requirements for 13C-NMR. For this journal, the evidence of 1H-NMR and HRMS spectrometry is generally enough, and we reviewed the ‘instructions for authors’ and found no relevant requirements for 13C-NMR. In addition, we are pleased to accept your suggestion and further supplement the 1H-NMR, 13C-NMR and HRMS spectrums of the important intermediate compound 3 (section 4.2.2; page 11) and the final product compound 6 (section 4.2.5; page 12).
Comment 3:
Scheme 1 should include yields, either within the graphic or else in the captions.
Response:
According to your comment, we added the yield information in the legend of Scheme 1 (page 3).
Comment 4: The previous work in the area (from the DiSimone group) is adequately cited. However, whereas the discussion does a nice job of explaining the similarities and differences between this work and the previous work (the previous work used silyl ethers as the direct precursors of active drug; this work uses the silyl ether as an acid-labile protecting element for a labile 4-hydroxybenzyl carbamate), the introduction (page 2, last paragraph) needs to be modified so that the reader will be aware that another group (DiSimone) looked at both trialkylsilyl and dialkyl silyl groups and, in the latter group, established the desirability of the diisopropylsilyl ethers employed here.
Response:
Thank you for your valuable advice. We have revised the introduction (page 2, last paragraph) to further clarify the differences between our research and previous work in this area (Finniss et al, Chem. Commun., 2014, 5,1355.). In previous work, the silyl ether groups were limited to conjutage hydroxyl-conjugated drugs. By introducing p-hydroxybenzyl alcohol (PHB), we achieves the efficient combination of amino-conjugated MMAE with acid-triggered silyl ether groups, and the resulting ADC exhibited significant efficacy in the activity evaluation.
Reviewer 4 Report
The manuscript by Wang et al suggests a new acid-cleavable silyl ether linker for ADCs. Overall, the data reported appear relatively complete, including demonstration of plasma stability, pH kinetics (from neutral to lysosomal pH), cytotoxicity, tubulin activity, cell cycle, in vivo efficacy, and some safety parameters. Also, the synthesis and characterization of LPs are mostly clear and complete.
From literature search, it appears that the silyl ether linker was previously proposed for ADCs by Finniss et al (Med Chem Commun, 2014), and this source is appropriately referenced in the manuscript as ref #38 in the results section. However it should be referenced in the introduction and/or conclusion to indicate that this linker system has been previously proposed for a Her2-ADC but that the current study adds PHB (unique from previous reports?) and expands on the biological utility.
In section 2.1, the authors state that different substitutions are not feasible due to different stabilities during synthesis, hence no SAR (structure-activity relationships) among the linker variants assessed. There should be reference to Supp Scheme S1 in this section. Also, the utility of the PHB moiety is not compared vs the absence of PHB.
The diisopropyl linker is quite hydrophobic and could impact PK; were other hydrophilic variants attempted to be synthesized?
The DAR of ~5.5 is higher than the typical ~4 reported for other MMAE ADCs. Is this the result of higher reduction, and were DAR 4 constructs effective, or is this linker more effective only in high DAR constructs (which could impact safety).
In Fig 2, the plasma stability of the ADC suggests ~24% loss over 7 days. This may be adequate for initial preclinical assessment but may need to be improved for therapeutic use of potent cytotoxins, including MMAE.
No PK data are shown, this is a standard approach to understand the quality of new ADCs in vivo.
The in vivo efficacy data in a single model with high Her2 expression (N87) is adequate. It would be useful to test the activity of a non-cleavable version of the linker to understand the impact of cleavage on activity. Also, at least one additional model (ideally moderate Her2 in different tumor type) would be beneficial to show the breadth of utility of this linker. It is understandable that such in vivo studies are labor intensive.
Other observations:
Table 1: Does the reported ADC IC50's represent nM Payload or Antibody? This should be indicated in a footnote. Also, the "mil40" header should state "Antibody mil40" for clarity. These are presumably nM Antibody concentrations and should also be listed in table footnote.
Fig 6: The doses reported are ng/ml; are these antibody or payload concentrations; please synchronize across the manuscript between the units and Payload/Antibody and indicate in text/tables/figures.
Section 2.10. Please indicate in text that the hematology was conducted day 28, 1 week after last dose (as indicated in methods).
Cytotoxicity assay: Cells were plated and then treatments added the same day. This is acceptable but without time for cells to adhere (typically 1 day), cell sensitivity to cytotoxics may be greater. OK as conducted and reported, but may produce lower IC50's than for other comparably reported studies.
This may be medium to high "Interest to Readers" who are attuned to new ADC linker approaches.
Some English grammar should be edited.
Overall, this is a reasonable report showing the utility of a new acid-cleavable linker system that may be applied to ADCs.
Author Response
Comment 1:
From literature search, it appears that the silyl ether linker was previously proposed for ADCs by Finniss et al (Med Chem Commun, 2014), and this source is appropriately referenced in the manuscript as ref #38 in the results section. However it should be referenced in the introduction and/or conclusion to indicate that this linker system has been previously proposed for a Her2-ADC but that the current study adds PHB (unique from previous reports?) and expands on the biological utility.
Response:
Thank you for your valuable advice. We revised the introduction (page 2, last paragraph) to further clarify the differences between our research and previous work, and cited the relevant reference (Chem. Commun., 2014, 5,1355.) in the introduction. We have added the following description to the introduction of the manuscript: “Although this type of linker has been tried for model ADC construction, which exhibits the characteristics of stability and acid-dependent cleavage, gemcitabine does not meet the high-efficiency requirements for ADC’s payload and it has not been evaluated for its efficacy.”
Comment 2:
In section 2.1, the authors state that different substitutions are not feasible due to different stabilities during synthesis, hence no SAR (structure-activity relationships) among the linker variants assessed. There should be reference to Supp Scheme S1 in this section. Also, the utility of the PHB moiety is not compared vs the absence of PHB.
Response:
Thank you for your thoughtful comment. In this study, we originally designed a variety of silyl ether-based linkers with different alkyl substituents. However, due to the instability or unreachability of some substances, we only get one target product, so we can’t perform SAR (structure-activity relationships) analysis. In our work, a novel acid-cleavage linker overcomes the application limitation of the silyl ether structure, which can be used for amino-conjugated cytotoxins by the introduction of PHB. When PHB is absent, the effective conjugating of MMAE with a silyl ether-based linker will not be achieved, and the traceless release of MMAE will not be achieved at the same time. Therefore, we are unable to compare its activity to the products without PHB.
Comment 3:
The diisopropyl linker is quite hydrophobic and could impact PK; were other hydrophilic variants attempted to be synthesized ?
Response:
Thank you for your advice. The hydrophilicity of the linkers is very important for ADCs because the increase in hydrophobicity will accelerate the elimination of the drug in vivo. In this study, the diisopropyl substituent is intended to increase the steric hindrance around the silicon atom, which is necessary to ensure the stability of the silyl ether. We have attempted to prepare a less hydrophobic silyl ether-based linkers that having a dimethyl or diethyl substituent, but it is unstable and not available. We will refer to your good suggestion, and try to further improve the hydrophilicity of this type of linkers by introduce a hydrophilic group into the spacer of the linkers in our future research.
Comment 4:
The DAR of ~5.5 is higher than the typical ~4 reported for other MMAE ADCs. Is this the result of higher reduction, and were DAR 4 constructs effective, or is this linker more effective only in high DAR constructs (which could impact safety).
Response:
Thank you for your valuable comment, and we agree with you that the DAR of ~5.5 may be higher than the typical value. The focus of our research is mainly on verifying the feasibility of this novel silyl ether-based acid-cleavable linker, which has been initially confirmed. In general, ADCs can be allowed for more payloads when its hydrophilicity is not significantly reduced (Nat Biotechnol. 2015, 33(7): 733-735.). For example, DS-8201 and IMMU-132, which have recently achieved significant clinical advances, have DAR values as high as 7~8. We observed that the ADC we prepared still has a low aggregation (~1%) even with higher loading ratio (DAR~5.5), so we used this batch of products in the next studies. In our future work, we will optimize the DAR properties of this type of ADCs according to your advice.
Comment 5:
In Fig 2, the plasma stability of the ADC suggests ~24% loss over 7 days. This may be adequate for initial preclinical assessment but may need to be improved for therapeutic use of potent cytotoxins, including MMAE.
Response:
Thank you for your kind advice. Objectively speaking, the silyl ether-based linker has the potential to further enhance the stability of acid-cleavable ADCs, and there is indeed a lot of work to be done in the future. More structural optimization work is expected to further improve the efficacy of this type of acid-cleavable ADCs.
Comment 6:
No PK data are shown, this is a standard approach to understand the quality of new ADCs in vivo.
Response:
Thank you for your professional advice. This study provides a series of indirect evidence from various perspectives such as plasma stability, efficacy, and safety studies, indicating that the resulting ADC has a similar half-life as conventional ADCs. We hope that these data could provide basic support for the initial feasibility of the novel acid-cleavable ADC, which is the main purpose of our research. The study of PK may be more meaningful for assessing the pharmaceutical properties of preferred compounds, which is a key task for us in the future.
Comment 7:
The in vivo efficacy data in a single model with high Her2 expression (N87) is adequate. It would be useful to test the activity of a non-cleavable version of the linker to understand the impact of cleavage on activity. Also, at least one additional model (ideally moderate Her2 in different tumor type) would be beneficial to show the breadth of utility of this linker. It is understandable that such in vivo studies are labor intensive.
Response:
The purpose and focus of this study was to develop a novel acid-cleavable linker to further expand its chemical space. Our research initially confirms the feasibility of this design concept. According to your suggestion, we hope to further carry out structural optimization work in the future and conduct further comparative studies on the best compounds to clarify the advantages of this design, which is the focus of future research. It is really difficult for us to complete the pharmacodynamic studies of other in vivo models in a limited time (10 days). Thank you again for your good advice and understanding.
Comment 8:
Table 1: Does the reported ADC IC50's represent nM Payload or Antibody? This should be indicated in a footnote. Also, the "mil40" header should state "Antibody mil40" for clarity. These are presumably nM Antibody concentrations and should also be listed in table footnote.
Response:
We further elaborate on the description of ADC’s concentration in Table 1 (page 5). The concentration of the ADC is counted based on the concentration of the antibody it contains.
Comment 9:
Fig 6: The doses reported are ng/ml; are these antibody or payload concentrations; please synchronize across the manuscript between the units and Payload/Antibody and indicate in text/tables/figures.
Response:
We have added an accurate description of the ADC concentration unit (ng/mL) in Figure 6 (section 2.8; page 7). Since the antibodies account for most of the mass in the ADC molecule (> 95%), the concentration of the ADC in this study was calculated as the corresponding antibody concentration for ease of measurement and comparison with the antibody.
Comment 10:
Section 2.10. Please indicate in text that the hematology was conducted day 28, 1 week after last dose (as indicated in methods).
Response:
We have made relevant adjustments to the manuscript (section 2.10; page 8), and described the time for the hematology analysis (after 1 week of stopping treatment).
Comment 11:
Cytotoxicity assay: Cells were plated and then treatments added the same day. This is acceptable but without time for cells to adhere (typically 1 day), cell sensitivity to cytotoxics may be greater. OK as conducted and reported, but may produce lower IC50's than for other comparably reported studies.
Response:
We are grateful to you for pointing out our error. The method of culturing the cells was incorrectly stated in the original manuscript (section 4.6; page 13), which has been corrected. Usually, we will incubate the cells at 37 ° C overnight before adding the test drug.
Comment 12:
This may be medium to high "Interest to Readers" who are attuned to new ADC linker approaches.
Response:
The silyl ether-based linker has great potential to improve the stability of acid-cleavable ADCs, and we will conduct further research according to your valuable suggestions.
Comment 13:
Some English gramma should be edited.
Response:
We sincerely apologize for the grammatical issues in our manuscript. Further, we have tried our best to make a comprehensive inspection and adjustment of the manuscript according to your suggestion.
Comment 14:
Overall, this is a reasonable report showing the utility of a new acid-cleavable linker system that may be applied to ADCs.
Response:
Thank you very much for your recognition of our work. According to your good suggestions, we will further modify the hydrophilicity of the linker, and carry out more in-depth pharmacodynamics and pharmacokinetic studies in the future work.
Once again, we appreciate for your warm work earnestly, thank you very much for your comments and suggestions. We will try our best to made some changes to the manuscript, and hope that the correction will meet with your approval.

Round 2
Reviewer 2 Report
The authors have not appropriately addressed comment 2. In addition, regarding comment 3 (ii), the authors should point to references, which describe similar efficacy of Herceptin and Herceptin-based ADCs to support their conclusions regarding similar inhibition of BT474 with mil40 and ADC. In this context, Lewis Phillips GD, et al (Cancer Research, 2008) shows that Trastuzumab is is very less effective than Trastuzumab-DM1 in BT474 cells.
Author Response
Comment 1:
The authors have not appropriately addressed comment 2.
Response:
Thank you again for your professional comment. This is indeed a pity for failing to use the same variable to study the effects of serum and pH in Figure 2 and Figure 3. In general, both released drugs and DAR changes can be used as effective indicators for assessing the performance of linker. In Figure 2, we initially selected MMAE as the detection indicator because it is a co-release substance of ADC and NAC-linker-MMAE. In Figure 3, we use DAR as a detection indicator to more intuitively reflect the real-time status of the complete ADC, and it would be better if the same indicators as in Figure 2 could be performed at the same time. According to your valuable suggestion, we will test these two indicators in future research to fully demonstrate the results.
Comment 2:
In addition, regarding comment 3 (ii), the authors should point to references, which describe similar efficacy of Herceptin and Herceptin-based ADCs to support their conclusions regarding similar inhibition of BT474 with mil40 and ADC. In this context, Lewis Phillips GD, et al (Cancer Research, 2008) shows that Trastuzumab is is very less effective than Trastuzumab-DM1 in BT474 cells.
Response:
Thank you very much for your valuable advice. 1) We added descriptions in the manuscript that describes the similar inhibition of antibody and ADC in BT-474 cells, and supplemented the relevant references (Int. J. Mol. Sci., 2017, 18, 1860.) to the corresponding locations in the manuscript (section 2.5; page 5). 2) ADCs typically exhibit superior inhibition rates over naked antibodies, and two other HER2-positive tumor cell lines (NCI-N87 and MDA-MB-453) in our study also supported this conclusion. For MMAE-based ADCs, there have been similar cases in which the maximum inhibition rates of naked antibody and ADC were similar in BT-474 cells (E.g: Breast Cancer Res Treat, 2015, 153(1):123-133.), which may be caused by different experimental conditions or some uncontrollable factors.
